# SimNet: Enabling Robust Unknown Object Manipulation from Pure Synthetic Data via Stereo

**Thomas Kollar**[*,1] **Michael Laskey**[*,1] **Kevin Stone**[*,1] **Brijen Thananjeyan**[*,1,2] **Mark Tjersland**[*,1]

**Abstract:** Robot manipulation of unknown objects in unstructured environments is a challenging problem due to the variety of shapes, materials, arrangements and lighting conditions. Even with large-scale real-world data collection, robust perception and manipulation of transparent and reflective objects across various lighting conditions remains challenging. To address these challenges we propose an approach to performing sim-to-real transfer of robotic perception. The underlying model, SimNet, is trained as a single multi-headed neural network using simulated stereo data as input and simulated object segmentation masks, 3D oriented bounding boxes (OBBs), object keypoints and disparity as output. A key component of SimNet is the incorporation of a learned stereo sub-network that predicts disparity. SimNet is evaluated on unknown object detection and deformable object keypoint detection and significantly outperforms a baseline that uses a structured light RGB-D sensor. By inferring grasp positions using the OBB and keypoint predictions, SimNet can be used to perform end-to-end manipulation of unknown objects across our fleet of Toyota HSR robots. In object grasping experiments, SimNet significantly outperforms the RBG-D baseline on optically challenging objects, suggesting that SimNet can enable robust manipulation of unknown objects, including transparent objects, in novel environments. Additional visualizations and materials are located at `https://tinyurl.com/simnet-corl`.

**Keywords:** Sim-to-Real, Computer Vision, Manipulation

## 1 Introduction

To deploy robots into diverse, unstructured environments, robots must have robust and general behaviors. Enabling general behaviors in complex environments, such as the home, requires robots to be able to perceive and manipulate previously unseen objects, such as new glass cups or t-shirts, even in the presence of variations in lighting, furniture, and objects. A promising approach to enable robust, generalized behaviors is to procedurally generate and automatically label large-scale datasets in simulation and use these datasets to train perception models [1, 2, 3, 4, 5, 6, 7, 8, 9, 10]. These models can extract the necessary representations for a wide variety of manipulation behaviors and can enable the manipulation policy to be implemented as a classical planner [2, 8, 11, 12, 13].

However, perception models trained purely on simulated RGB data can overfit to simulation artifacts such as texture and lighting. In order to explicitly force models to focus on geometric features instead, models are often trained on active depth information [1, 2, 3, 4, 6]. However, active depth sensors use structured light, which struggles in environments where reflective and transparent objects are present [3]. Natural home environments, which often have harsh lighting conditions and reflective or transparent objects such as glassware, motivates designing a method that is robust to these variations and can leverage geometric features without using depth sensors.

An alternative to active depth sensing is passive stereo matching, which captures images from two cameras and matches pixels in each image to a single point in 3D space. The disparity, or horizontal difference in the pixel coordinates, of the point can be directly mapped to depth [14]. Recent work from the stereo vision community shows how to approximately perform stereo matching to predict depth images using a differentiable cost volume neural network that matches features in stereo images [15, 16]. Unlike prior work, which typically focuses on accurate reconstruction of depth images, in this work the "low-level" features from approximate stereo matching [5, 17] are used as an intermediate representation for "high-level" vision tasks.

[1]Toyota Research Institute. [2]University of California, Berkeley.
[*] Denotes equal contribution, $\alpha - \beta$ ordering.
Contact: {first name.last name} @ tri.global

5th Conference on Robot Learning (CoRL 2021), London, UK.

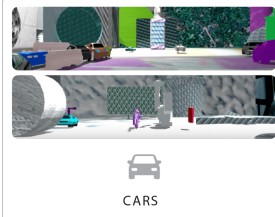 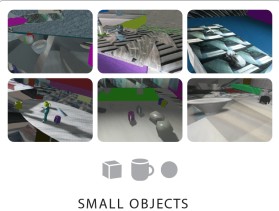 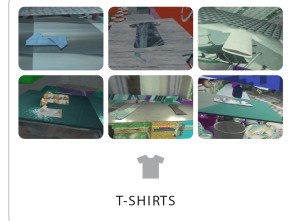

CARS      SMALL OBJECTS      T-SHIRTS

Figure 1: **Simulation images:** Simulated data was generated for three domains: cars, small objects and t-shirts using a non-photorealistic simulator with domain-randomization. Dataset generation is parallelized across machines and can be generated in an hour for $60 (USD) cloud compute cost. By forcing the network to learn geometric features, sim-to-real transfer can be performed using only very low-quality scenes.

To make perception robust in challenging home environments, we introduce SimNet, a lightweight neural network which leverages "low-level" vision features from a learned stereo network for "high-level" vision tasks (Fig. 2). SimNet is able to be trained entirely on simulated data. By forcing the network to focus on geometric features using domain-randomized data (Fig. 1) and by using a learned stereo network that is robust to diverse environments, SimNet learns to robustly predict representations that can be used for manipulation of unknown objects in novel scenes. SimNet is able to predict a variety of "high-level" outputs, including segmentation masks, 3D oriented bounding boxes and keypoints. In contrast to previous work on manipulating unknown objects in novel environments, SimNet does not need large-scale real data collection [3, 18, 19, 20], active depth sensing [1, 3, 4], or photorealistic simulation [3, 21].

SimNet has been evaluated on three real-world domains: unknown object grasping, t-shirt folding, and 2D car detection. For unknown object grasping, SimNet shows accuracy gains over previous depth-based approaches and is able to grasp optically challenging transparent objects even where other depth-based approaches fail (Fig. 3). The demonstration of t-shirt folding shows that sim-to-real transfer is possible with SimNet even in the presence of deformable objects and challenging optical conditions. Finally, ablations of the SimNet model components are performed on the KITTI [22] 2D car detection task. These experiments demonstrate the benefits of using a cost-volume for stereo matching to enable sim-to-real transfer.

This work makes three contributions: *(i)* an efficient neural network for sim-to-real transfer, SimNet, that uses learned stereo matching to enable robust sim-to-real transfer of "high-level" vision tasks, such as keypoints and oriented bounding boxes (OBBs), *(ii)* the first network to enable direct prediction of 3D oriented bounding boxes of unknown objects *(iii)* and an indoor scenes dataset with 3D oriented bounding box labels of common household objects, corresponding stereo and RGB-D images, and training code for our model.

## 2 Related Work

The related work is described for perception for manipulation, sim-to-real transfer and learned stereo.

### 2.1 Parameterized Representations for Manipulation

In contrast to end-to-end robot policy learning [19, 23, 24, 25], a popular approach for generating behaviors is to parameterize motions by the outputs of a perception network [1, 2, 8, 11, 26]. This decouples perception from planning and control, and enables perception systems to be trained in simulation without the need for accurate physical simulation. One representation commonly used for grasping of rigid objects are oriented bounding boxes (OBBs) [26]. By aligning grasps along the principal components, OBBs can be used to grasp common household objects [26]. Our work contributes a method to directly predict OBBs from a neural network. A representation that is used for deformable manipulation are keypoints or learned correspondences, which have seen significant success in tasks such as deformable manipulation [2, 8, 10, 11, 12, 27] and grasping [28, 29]. Ganapathi et al. [8] and Sundaresan et al. [2] predict correspondences from domain-randomized depth or monocular RGB images and demonstrate impressive transfer to real fabrics and ropes in constrained lab environments. In contrast, we demonstrate robust transfer on diverse home scenes, and compare to the methods in these papers as baselines.

### 2.2 Sim-to-Real Transfer

Enabling perception models trained on simulated data to transfer to real-world scenes is an active area of research. One technique to transfer from simulation to reality is to train directly on a geometric

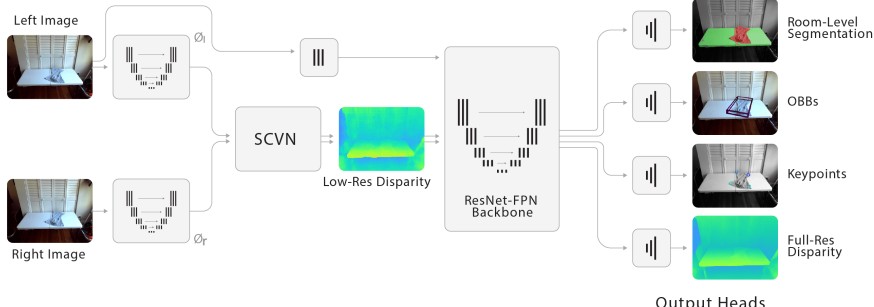

Figure 2: **SimNet:** In SimNet, each stereo RGB image is fed into a feature extractor before being fed into SCVN, which performs approximate stereo matching. The output of the SCVN is a low resolution disparity image fed in with features from the left image to a ResNet-FPN backbone and output prediction heads. The output heads predict room-level segmentation, OBBs, keypoints, and full-resolution disparity images.

representation of the world such as a point cloud or depth image [1, 2]. Prior work has used depth-based transfer techniques in a variety of scenarios, such as grasping unknown objects [1], tying a rope [2], and segmenting object instances in cluttered scenes [4, 6]. Depth-based transfer is commonly performed with an active sensor using structured light. The applicability of these sensors is limited to regimes where their reflected infrared pattern is consistent enough that it can be pattern matched for 3-D reconstruction [30]. However, home settings contain non-Lambertian objects such as glassware and a large amount of natural light, which can create large sensing errors. Existing work fills in missing depth information for transparent objects by combining depth and RGB information from photorealistic simulation data and real data [3]. In contrast to [3], which uses photorealistic simulation that requires artists to create high quality assets, SimNet uses inexpensive RGB sensors and low quality simulated data for training. SimNet is comparable to Xie et al. [4], which separately leverages depth and synthetic RGB inputs to compensate for the limitations of depth sensing alone. SimNet is compared to several variants of Xie et al. [4], and is found to outperform them all.

An alternative to depth-based transfer is domain randomization (DR) with RGB images [5, 17, 31, 32]. Typically, DR randomizes over the lighting and textures in the environment by utilizing a low-quality, non-photorealistic renderer. DR has produced successful results in car detection [31], 6-DOF pose estimation of known objects [33], and camera calibration [34]. By injecting large randomization into texture and lighting, the network is forced to use the geometry of the scene to solve the task [17]. A limitation of this approach is that geometric reasoning on a monocular image requires leveraging global shape priors, lighting effects, and scene understanding [35]. In this work, a sim-to-real transfer technique is proposed that uses learned stereo matching and domain randomization to extract geometric features using only local non-semantic information. Robustness of sim-to-real transfer is demonstrated on the KITTI benchmark. A similar result, [21], was published concurrently and also demonstrates the advantages of stereo for sim-to-real transfer when using photorealistic simulation.

### 2.3 Learned Stereo Matching

The goal of stereo matching is to explicitly compute the pixel displacement offset between objects from the left and right images in a stereo pair [14]. By computing a per-pixel displacement offset, or disparity, a high-resolution point cloud can be generated of the world. Given that it is challenging to estimate similarity features of image patches, recent work has explored using deep learning to advance stereo techniques [15, 16, 36, 37, 38, 39, 40]. In contrast to the learned stereo community that targets achieving perfect depth-sensing when training with real data [15, 16], SimNet uses depth-sensing to enable sim-to-real transfer for "high-level" vision tasks by regressing coarse depth with an efficient stereo matching network architecture based on [15, 16] and learning features that are robust enough to predict oriented bounding boxes, segmentation masks, and keypoints on real objects for manipulation.

## 3 SimNet: Enabling Predictions for Manipulation From Synthetic Stereo

In this section, we will discuss our network architecture that leverages approximate stereo matching techniques from [15, 16] and domain randomization [5, 17] to predict segmentation masks, OBBs, and keypoints on unseen objects for robot manipulation. Our key insight is that by training approximate stereo matching algorithms on pure synthetic data, robust "low-level" features like disparity can be learned, thereby enabling sim-to-real transfer on "high-level" vision tasks. First, we discuss how to learn robust low-level features, which are then used for "high-level" perception. Then, we discuss

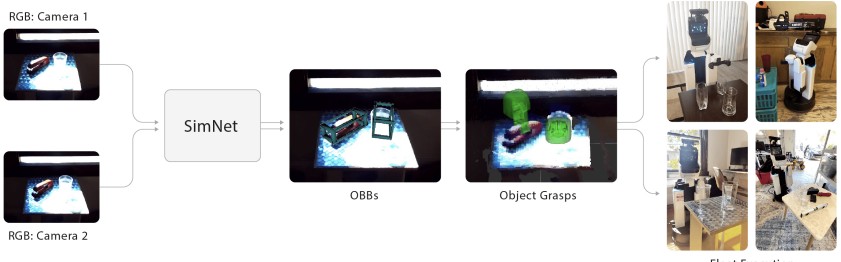

Figure 3: **Manipulation overview**: A stereo pair is fed as input to SimNet, which produces OBBs. Grasp positions are produced from OBBs and are used by a classical planner to grasp the object [26]. This grasping technique has been deployed on a fleet of home robots to perform manipulation in optically challenging scenarios.

how our low-cost synthetic data is generated. For training details see supplemental Sec. B.2; the overall network architecture is in Fig. 2.

### 3.1 Stereo Cost Volume Networks (SCVN) For Robust Low-Level Features

The stereo cost volume network (SCVN) for performing robust learned stereo matching is described in this section; ablations performed in Sec. 4.4 show its benefits. Let $\odot$ to denote Hadamard products, and $I_{[i,j:k,:]}$ denote the selection of all elements with index $i$ in the first dimension of tensor $I$, index in $\{j, \ldots k-1\}$ in the second dimension of $I$, and any index in the third dimension onwards. Let $I_l$ and $I_r$ denote the left and right RGB images from the stereo pair. Each image has dimension $3 \times H_0 \times W_0$. The left and right images are fed into neural networks $\Phi_l$ and $\Phi_r$ that featurize each image respectively and output feature volumes $\phi_l$ and $\phi_r$. Both $\phi_l$ and $\phi_r$ have dimension $C_\phi \times H_\phi \times W_\phi$, where $C_\phi$ is the number of channels in each feature volume, and $H_\phi$ and $W_\phi$ are their height and width, respectively. We used a lightweight Dilated ResNet-FPN [41] as our feature extractor, to enable large receptive fields with a minimal amount of convolutional layers.

The extracted features $\phi_l$ and $\phi_r$ are fed into a stereo cost volume network $f_{\text{cost}}$ that consists of an approximate stereo matching module that searches horizontally in the feature volumes for correspondences within an allowed disparity range. In classical stereo vision literature, correspondences across left and right images can be found by searching along a horizontal line across the images for a match, and the disparity is the difference in the $x$ coordinates in the match, which is high for closer points in 3D space and low for farther points. The architecture for $f_{\text{cost}}$ is heavily inspired by the techniques in [15] and approximately performs this search. The first phase of the network $f_{\text{cost}}^{(0)}$ computes pixelwise dot products between horizontally shifted versions of the feature volumes. The output of this phase has dimension $C_c \times H_\phi \times W_\phi$. The value $2*(C_c - 1)$ represents the maximum disparity considered by the network, and the minimum disparity considered is 0. The $i$-th $H_c \times W_c$ slice of the output is computed as:

$$f_{\text{cost}}^{(0)}(\phi_l, \phi_r)_{[i,:,i:]} = \sum_{j=0}^{C-1} \left( \phi_{l,[i,:,i:]} \odot \phi_{r,[i,:,:W-i]} \right)_{[j]}; \qquad f_{\text{cost}}^{(0)}(\phi_l, \phi_r)_{[i,:,:i]} = 0$$

The first case takes the rightmost $H_c - i$ columns of the left feature volume $\phi_l$ and computes a pixelwise dot product with the leftmost $H_c - i$ columns of $\phi_r$. This operation horizontally searches for matches across the two feature volumes at a disparity of $2i$. The next phase of the network $f_{\text{cost}}^{(1)}$ feeds the resulting volume into a sequence of ResNet blocks, which outputs a volume of dimension $C_c \times H_\phi \times W_\phi$ before performing a soft argmin along the first axis of the volume. The soft argmin operation approximately finds the disparity for each pixel by locating its best match. The final volume is an estimate of a low-resolution disparity image $\hat{I}_{\text{d,low}}$ with shape $H_\phi \times W_\phi$. We denote $f_{\text{cost}} = f_{\text{cost}}^{(1)} \circ f_{\text{cost}}^{(0)}$.

**Disparity Auxiliary Loss** In addition to the losses for the high-level perception heads (Sec. 3.2 and 3.3), the weights of $\Phi_l$, $\Phi_r$, and $f_{\text{cost}}$ are trained by minimizing an auxiliary depth reconstruction loss function. In particular, the loss function takes in a target disparity image $I_{\text{targ,d}}$ of dimension $H_0 \times W_0$, downsamples it by a factor of $H_0/H_\phi$ and then computes the Huber loss [42] $\ell_{\text{d,small}}$ of it

with the low-resolution depth prediction $f_{\text{cost}}(\phi_l, \phi_r)$. That is, the network weights are trained to minimize $\ell_{\text{d,small}}(f_{\text{cost}}(\phi_l, \phi_r), \texttt{downsample}(I_{\text{targ,d}}, H_0/H_\phi))$.

## 3.2 Extracting High-Level Predictions for Manipulation Tasks

Given a SCVN to extract geometric features from stereo images, we need to learn high-level predictions relevant to manipulation. To design a backbone for robust simulation-trained manipulation, we feed the output of a SVCN, $\hat{I}_{\text{d,low}}$, into a Resnet18-FPN [43] feature backbone $f_{\text{backbone}}$. Additionally, early stage features from the left RGB image, $I_l$, allow high resolution texture information to be considered at inference time, similar to [4]. The features are extracted from the ResNet stem, concatenated with the output of the SCVN and fed into the backbone. The output of the backbone is fed into each of the prediction heads.

The rest of this section describes how SimNet uses the output of the backbone for the core prediction heads (Fig. 2) and the losses used for training the network. The optional auxiliary prediction heads (i.e. segmentation and refined disparity) are described in Sec. 3.3. Each prediction head uses the up-scaling branch defined in [43], which aggregates different resolutions across the feature extractor.

**Oriented Bounding Boxes:** Detection of an OBB requires determining individual object instances as well as estimating translation, $t \in \mathbb{R}^3$, scale $S \in \mathbb{R}^{3\times3}$, and rotation, $R \in \mathbb{R}^{3\times3}$, of the encompassing OBB. These parameters can be recovered by using four different output heads. First, to recover object instances, we regress a $W_0 \times H_0$ image, which is the resolution of the input left image, where for each object in the image a Gaussian heatmap is predicted. Instances can then be derived using peak detection as in [44, 45]. We use an $L_1$ loss on this output head and denote the loss as $l_{\text{inst}}$.

Given instances of object, the remaining 9-DOF pose parameters can be regressed. To recover scale and translation, we first regress a $W_0/8 \times H_0/8 \times 16$ output head where each element contains pixel-wise offset from detected peak to the 8 box vertices projected on to the image. Scale and translation of the box can be recovered up to a scale ambiguity using EPnP similar to [44]. In contrast to prior work in pose estimation, the predicted box is aligned based on principal axes sized in a fixed reference frame. To recover absolute scale and translation, the distance from the camera $z \in \mathbb{R}$ of the box centroid is regressed as a $W_0/8 \times H_0/8$ tensor. The two losses on these tensors are an $L_1$ loss and are denoted $l_{\text{vrtx}}$ and $l_{\text{cent}}$.

Finally, the rotation of the OBB, $R$, can be recovered via directly predicting the covariance matrix, $\Sigma \in \mathbb{R}^{3\times3}$ of the ground truth 3D point cloud of the target object, which can be easily generated in simulation. The output tensor of $W_0/8 \times H_0/8 \times 6$ is directly regressed, where each pixel contains both the diagonal and symmetric off diagonal elements of the target covariance matrix. Rotation can then be recovered based on the SVD of $\Sigma$. $L_1$ loss on this output head is used and denoted as $l_{\text{cov}}$. Note that for the 9-DOF pose loses, the loss is only enforced when the Gaussian heatmaps have score greater than $0.3$ to prevent ambiguity in empty space. For more details on the OBB prediction head, the code is located here: https://github.com/ToyotaResearchInstitute/simnet.

**Keypoints:** Keypoints and learned correspondences are a common representation for robot manipulation, especially in deformable manipulation [2, 8, 11, 12, 28]. SimNet has an output head that predicts keypoints, such as t-shirt sleeves for t-shirt folding (Sec. 4). The output head predicts heatmaps for each keypoint class, and is trained to match target heatmaps with Gaussian distributions placed at each ground-truth keypoint location using a pixelwise cross-entropy loss $l_{\text{kp}}$. To extract keypoints from the predicted heatmaps, non-maximum suppression is used to perform peak detection [44].

## 3.3 High-level Predictions: Optional Auxiliary Prediction Heads

Two optional auxiliary prediction heads can be used to enable better scene understanding of the world. These prediction heads do not affect performance of the other tasks.

**Room Level Segmentation:** A room-level segmentation head can predict one of three categories: surfaces, objects and background. Cross-entropy loss $l_{\text{seg}}$ is used for training, as in prior work [43]. This prediction head enables a mobile robot to detect surfaces and manipulable objects.

**Full Resolution Disparity:** Since the SCVN produces a disparity image at quarter resolution, the feature extractor can combine the backbone and the left stereo image to produce a full resolution depth image. The same branch architecture as the previous heads is used to aggregate information across different scales. During training we use the same loss as the SCVN, but enforced at full resolution. This output is trained using a Huber loss function and is denoted $\ell_{\text{d}}$. A full resolution disparity image can be converted into a 3D point cloud for collision avoidance.

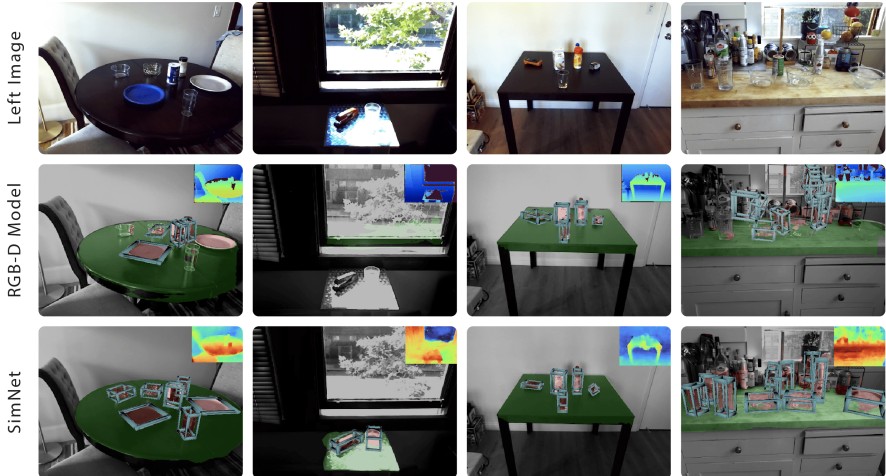

Figure 4: **Graspable Object Predictions:** SimNet is evaluated on an OBB regression task with objects of varying sizes and shapes on flat surfaces. Top: The left RGB image from different homes. Middle: Our model prediction trained with RGB-D sim-to-real transfer similar to [4]. The top right corner is the output of the Asus Xtion depth sensor. Bottom: SimNet. In the top right, we show the low-res disparity estimate predicted by the learned stereo net. SimNet consistently enables better sim2real transfer of the predictions for optically challenging scenarios.

### 3.4 Efficient Synthetic Dataset Generation

Given the complexity of all the output predictions defined in the previous section, it would be impractical to label a sufficient amount of real data to generalize across scenes. Thus, we are interested in using synthetic data to provide ground truth annotations on a wide variety of scenarios. To force the network to learn geometric features, we randomize over lighting and textures as recommended in Mayer et al. [38]. In contrast to [3, 46], we use OpenGL shaders with PyRender [47] instead of physically based rendering [48] approaches. Low-quality rendering greatly speeds up computation, and allows for dataset generation on the order of an hour. Three datasets are generated: cars, graspable objects, and t-shirts (Fig. 1). Additional details about the datasets are in supplemental Sec. D. The training and validation data for the graspable object dataset can be found here: `https://github.com/ToyotaResearchInstitute/simnet`.

## 4 Experiments

Three real-world computer vision and robotics experiments are performed to evaluate how well SimNet can learn from synthetic stereo data and transfer to diverse, real images in unstructured environments. First, SimNet is compared to baseline approaches on the small objects dataset and an ablation of multi-task training is performed to understand its benefits. Second, the best model from the first experiment is used to evaluate SimNet's performance on two optically challenging manipulation tasks: (1) grasping unknown objects on tabletops and (2) t-shirt folding. Third, sim-to-real transfer for new domains is evaluated using the KITTI 2-D car detection task; additional ablations are performed to show the benefit of the SCVN. With the exception of the manipulation experiments on the robot, all of the perception results are averaged over 3 trials; little variation between trials is noted.

All physical experiments are conducted on tabletops found across four different, real homes. Manipulation is performed with the Toyota HSR robot, which has a four DOF arm, mobile base, and pair of parallel jaw grippers [49]. It has a mounted stereo pair from a Zed 2 camera and a Asus Xtion Pro RGB-D sensor. Each home has different background objects, furniture, graspable objects, and lighting conditions, which evaluates each network's ability to robustly generalize to diverse scenarios.

### 4.1 Ablation of Baselines on Validation Data

In this experiment, SimNet is trained on the small objects dataset and evaluated on a manually annotated real-world validation set. SimNet is compared to the following baselines: **mono**, which uses the left stereo image, **depth** [50], which only uses depth inputs, and **RGB-D**, which uses both RGB and depth inputs. To make the RGB-D baseline competitive, the best among four different

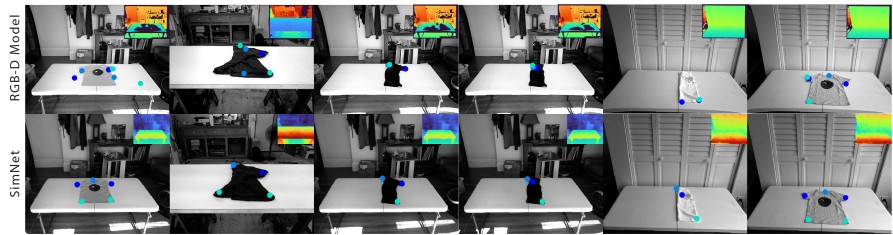

Figure 5: **T-shirt Keypoint Predictions:** SimNet is evaluated on keypoint regression for shirts in various stages of folding. Three classes of keypoints are predicted: sleeves, neck, and bottom corners. RGB-D performs poorly due to strong natural lighting and minimal depth variation. SimNet accurately predicts keypoints on the shirts despite these challenges.

approaches to fuse color and depth information was picked (See RGB-D baselines in Appendix, Section B.1) by evaluating on a held out set of 500 non-optically challenging scenes (i.e. matte objects with minimal natural light). SimNet is compared to a technique inspired by [4], **RGB-D Seq**, where inference is done in a two-stage manner first on depth and then on RGB (see Section B.1). The performance of SimNet after turning off all auxiliary tasks (except auxilliary depth loss) is reported as **SimNet- w/o Aux.**. 3D mAP@0.25 of annotated bounding boxes of each object, a common metric in pose prediction [51], was used as an evaluation metric (see Appendix B.3). Results are shown in Table 1. Additional sensing ablations are in the Appendix, Sec. C.

Monocular sensing performs poorly on this task, likely due to the task being 3D in nature, while depth and RGB-D perform much better. Both versions of **SimNet** (with or without auxiliary tasks) perform better than all other sim-to-real techniques, including using RGB-D and the approach inspired by [4] (**RGB-D Seq**). The similar mAP score between **SimNet-w/o Aux.** tasks and **SimNet** shows that the additional output heads are optional and not needed for sim-to-real transfer. These results indicate that stereo matching can lead to more robust sim-to-real transfer.

| Method | Mono [5] | Depth [50] | RGB-D | RGB-D Seq [4] | SimNet | SimNet- w/o Aux. |
|--------|----------|------------|-------|---------------|--------|------------------|
| mAP    | 0.164    | 0.831      | 0.855 | 0.774         | 0.921  | 0.947            |

Table 1: **SimNet Perception Results:** An ablation of different sensing modalities on the 3D OBB prediction task evaluated using 3D mAP on a dataset of real, human-annotated images of optically **easy objects.**

## 4.2 Unknown Object Grasping

In this manipulation experiment, the robot's task is to grasp objects on a tabletop comprised of two classes of household objects in each home: **optically easy**, which consists of opaque, non-reflective objects, and **optically hard**, which contains optically challenging, transparent objects (Fig. 6). For each trial, an object is selected uniformly at random from the dataset and randomly placed on the tabletop with other distractor objects. The task is to grasp the foremost object in the scene using a heuristic grasp planner that takes OBBs as input (Fig. 3). To grasp objects the robot aligns the gripper with the largest principal axis of the OBB. In the event of similar sized principal axes like a ball, the robot favors grasping the object on the side closest to the robot. A grasp is successful if the robot is able to raise the object off the table and remove it from the scene.

For each of the four homes we test five easy objects and five hard objects and compare SimNet against the best RGB-D baseline found in Sec. 4.1. Quantitative results are reported in Table 2 and qualitative results of the predictions are in Fig. 4. The results show that SimNet outperforms RGB-D, 92.5% vs. 62.5% in grasp success. On optically easy objects RGBD-D and SimNet perform similarly on average, while SimNet performs much better when grasping challenging objects such as glassware. These results demonstrate that SimNet is able to robustly grasp a variety of unknown challenging objects, even in optically challenging settings.

## 4.3 Grasp Point Prediction in T-shirt Folding

In this manipulation experiment, the robot is evaluated on a t-shirt folding task, where it must execute a sequence of four folds on unseen, real t-shirts. This task is challenging to perform using depth sensing, because the depth resolution of most commercial depth sensors cannot capture the subtle variations in depth due to the thickness of a t-shirt. Keypoints are a popular representation for

| Method (Object Class) | Home 1 | Home 2 | Home 3 | Home 4 | Overall |
|---|---|---|---|---|---|
| RGB-D (O. Easy) | 4/5 | **5/5** | 4/5 | **5/5** | **18/20** |
| SimNet (O. Easy) | **5/5** | 4/5 | **5/5** | 4/5 | **18/20** |
| RGB-D (O. Hard) | 0/5 | 1/5 | 1/5 | **5/5** | 7/20 |
| SimNet (O. Hard) | **5/5** | **5/5** | **5/5** | 4/5 | **19/20** |

Table 2: **SimNet Grasping Results:** Grasp success scores comparing the best RGB-D method and SimNet.

manipulating deformable objects [11, 12]. A shirt folding policy is parameterized using keypoint predictions for the shirt's neck, sleeves, and bottom corners (Sec. B.4.2). To compute quantitative results on sim2real transfer, a validation dataset of 32 real images is collected from 12 t-shirts in 3 homes (Fig. 5). Keypoint prediction mAP is reported (Table 3). Videos of t-shirt folding are presented on the project website: `https://tinyurl.com/simnet-corl`.

The results show that SimNet significantly outperforms RGB-D and depth, and slightly outperforms mono. This is likely due to interference from natural lighting and low depth profile of the shirts. The monocular network transfers well to the real t-shirt images, and SimNet slightly outperforms it. Unlike the models that use active depth sensing, SimNet is robust to lighting variation and the low depth variation of the shirts. Also, these experiments indicate that SimNet can accurately predict object keypoints, thereby enabling manipulation of certain deformable objects.

| Method | Mono | Depth | RGB-D | SimNet |
|---|---|---|---|---|
| mAP | 0.893 | 0.282 | 0.631 | **0.917** |

Table 3: **T-Shirt Folding Results:** A comparison of t-shirt keypoint mAP across different sim-to-real approaches.

## 4.4   SCVN Ablation on KITTI 2-D Car Detection

To motivate the use of the SCVN for achieving sim-to-real transfer, evaluation on the standard 2-D car detection benchmark on KITTI was performed [22] since it has been used as a challenging sim-to-real domain in prior work [31]. The models compared include SimNet (with the SCVN), which uses simulated stereo images, **Mono**, which uses monocular inputs, **Stacked**, which uses stacked left and right image inputs, **SimNet-No Depth Loss**, which uses synthetic stereo images but does not have a depth-auxiliary loss on the output of the SCVN and **SimNet-real**, which uses real training data with a standard validation/training split used in [45]. To predict 2-D bounding boxes, SimNet is modified to use the single-shot box detection output head in CenterNet [45]. mAP@0.5 is reported, similar to [31], for the aggregate of moderate and easy car classes.

The results can be seen in Table 4. They show that SimNet can significantly outperform monocular images and naive stereo concatenation, which suggests explicit stereo matching leads to more robust transfer. However, if no auxiliary depth loss is enforced, there is a significant decrease in performance, indicating that the SCVN is very important for sim-to-real transfer. Furthermore, the gap between real and sim data is only 3.5%, which suggests relatively robust sim-to-real transfer. However, qualitatively SimNet fails to detect cars at distance, since the disparity for their pixels becomes very small as distance from the camera becomes large. Visualizations of the predictions can be seen the Appendix, Fig. 13. Thus, SimNet is best used for applications with limited range from the camera, such as for robot manipulation.

| Method | Mono | Stacked | SimNet-No Depth Loss | SimNet | SimNet-real |
|---|---|---|---|---|---|
| mAP | 0.565 | 0.710 | 0.612 | **0.826** | 0.861 |

Table 4: **KITTI Results:** Comparison of different sim-to-real techniques for 2D car detection on KITTI.

## 5   Conclusion

We present SimNet, an efficient, multi-headed prediction network that leverages approximate stereo matching to transfer from simulation to reality. The network is trained entirely on simulated data and robustly transfers to real images of unknown optically-challenging objects such as glassware, even in direct sunlight. We show that these predictions are sufficient for robot manipulation such as t-shirt folding and grasping. In future work, we plan to use SimNet for enabling 3D scene understanding in homes.

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
