# OpenReview forum: "SimNet: Enabling Robust Unknown Object Manipulation from Pure Synthetic Data via Stereo"
_robot-learning.org/CoRL/2021/Conference — CoRL2021 Poster_

### Official Review · Reviewer_m12X · 2021-07-12

**Originality:** Very Good
**Technical Quality:** Good
**Clarity Of Presentation:** Good
**Impact:** 4

**Recommendation:**

Weak Accept: I recommend accepting the paper, but will not argue for my recommendation if the majority of other reviewers have a different opinion.

**Summary:**

This paper proposes an approach for sim2real transfer that can leverage suboptimal simulation data for downstream robotic manipulation tasks on unknown objects. The approach is based on a deep neural network that is trained on jointly optimizing a variety of perception-based losses from stereo images. To reduce the simulation gap, the model is trained on predicting an auxiliary disparsity map between its input images, which is then used as additional information for the later predictions. This incentivizes the model to focus on geometric reasoning rather than texture or depth. Experimentally, the authors show that this geometric reasoning improves downstream performance on object detection and manipulation tasks when compared to similar models that are trained without it.

**Issues:**

- The baseline in Section 4.1.2. is chosen by "sweeping over four different algorithms", which are not explained further in the main paper.
- The experimental results are presented as plain numbers without standard deviations, confidence intervals or other measures of uncertainty. This makes it hard to assess the stability of the approach as well as the qualitative difference between the approach and its baselines.
- Section 4.4 reports mAP@ 0.5, while previous sections report mAP@ 0.25. This choice is not further explained/justified in the paper. Either making this consistent or explaining the reason for this change would help the experimental section.
- The appendix mentions that the weights for the losses for the different models are chosen based on HyperBand. This should be part of the main paper. It should also be clarified how the other hyperparameters were chosen.
- Contributions **(iii)** to **(v)** each refer to one experimental setup and the insights taken from it. This is more of a description and evaluation rather than a contribution. The insights (better transfer than baselines, useful for downstream policies) could instead be added mentioned in contribution **(i).**
- The RGB-D baseline in Section 4.1.2 is chosen based on the performance for "easy" objects. Later on, this baseline performs bad on the "hard" objects, which could be an artefact of architecturally overfitting on the "easy" objects. The bad baseline performance (or rather the comparison of the model's performance to it) is a main selling point of the paper and highlighted in the abstract. As such, the baseline should be convincingly designed to be as strong as possible.
- The car detection (Section 4.4) uses a "Stacked" baseline consisting of stereo images that is missing in previous sections. Considering the good performance of the "Mono" baseline in the previous section, it would be interesting to see how "Stacked" compares to the main model on the other tasks.
- Some heads of the model use an L1 loss, others use a Huber loss. It is not clear why this distinction is made and why which loss is chosen.

### Minor Issues

- Table 1 shows the Perception performance of the model compared to baselines for "easy" objects. It would be interesting to also see the performance for "hard" objects.
- The name *SimNet* is already used by NVIDIA for a simulation toolkit
- The appendix is very bloated due to the figures having high resolution. The file itself is rather big, and some page takes a second to load
- The model is presented as an end-to-end method in the appendix and the introduction. While one can argue that is is trained in an end-to-end fashion, this is different from end-to-end methods for e.g., robot policy learning. This distinction is briefly mentioned in the related work section, but not really cleared up.
- The model feeds early features from the left image to its main backbone, but does not do so for the right image. This is presumably done because the left image is the basis for the output heads, but that is not explicitly mentioned in the paper.

### Typos and Style

- The model concept is explicitly described/explained in the abstract and each of the early sections. This feels somewhat repetitive and could be alleviated by shorter or more targeted descriptions.
- Ll. 9, 10: The appendix mentions "disparsity" twice, once for the final output and once for the auxiliary loss. This distinction is not clear.
- Ll. 29, 41: References to figures that are very far away from the current text
- Ll. 35: "artefacts" → "artifacts"
- Ll. 69: Contribution (ii) explicitly mentions single-shot prediction, but the rest of the paper mostly does so implicitly. If this is a selling point, it should maybe be more prominent in the paper.
- Ll. 142: The double-subscript Notation for the images is a little cluttered. Maybe the image position (*l, r*) could instead be indicated via a superscript.
- Ll. 158: Consecutive sentences start and end with an equation, which makes this hard to read.
- Ll. 201: EPnP is used as an acronym but never written out.
- Ll. 210: SVD is written in bold while other methods and/or acronyms are not.
- Ll. 263: "optically easy" and "optically hard" should be made consistent with other parts of the paper, where the distinction is simply made between "easy" and "hard" objects.
- Ll. 298: The reasoning "since the ability..." is stated as a fact, and should be supported by citations or other evidence.
- Ll. 302: "am" → "an"

**Reviewer Expertise:**

Good: General knowledge of the area

**Strengths And Weaknesses:**

### Strengths

- The main idea of training an auxiliary disparsity map from stereo inputs is a conceptually simple but effective approach to utilize geometric information in sim2real transfer.
- This allows for good performance on downstream real-world tasks even when the model is trained on sub-optimal simulations, thus mitigating the need for slow and expensive hyper-realistic simulations.
- The approach is applied to different real-world tasks, showcasing its versatility.
- Both the related work and the model description are presented clearly. The figures and schematics are very fitting. The model is concisely summarized in the schematic in Figure 2, and Figure 4 shows a conclusive quantitative example of the advantages of the approach.

### Weaknesses

- The model depends on a plethora of different training objectives. As such, it needs to tune a weight for each of those, making it less applicable in novel scenarios. Adding up a number of losses like this also comes with the risk of the network prioritizing some losses and ignoring others, thus potentially being less reliable if not carefully tuned.
- On the same note, the impact of the different training losses for the final performance is not really explored beyond the auxiliary loss. An additional ablation study on these losses would be useful.

**Summary Of Recommendation:**

The paper presents and explores the interesting idea of incentivizing geometric reasoning in a neural architecture by means of an auxiliary disparsity loss to improve sim2real transfer. The individual parts of the rather complex model are both intuitive and motivated nicely.

The approach is evaluated on different real-world scenarios and shown to achieve good performance. However, the experiments could be more convincing as they lack uncertainty measures (such as standard deviations of the performance over multiple trials) and some insightful ablations. It can be argued that the baselines that the method is compared to could have been stronger.


========================================

(Edited and copied from the reviewer response below)
The authors have resolved many of my concerns, while indicating that they intend to address most of the remaining ones if accepted. I can not evaluate intended changes, but still believe that the paper in its current state poses an interesting contribution. I will leave my score at `weak accept'.

---

> ### Author Response · Authors · 2021-08-25
> **Clarification of Experiments and Multi-task Learning Objective**
>
> >> The model depends on a plethora of different training objectives. As such, it needs to tune a weight for each of those, making it less applicable in novel scenarios. Adding up a number of losses like this also comes with the risk of the network prioritizing some losses and ignoring others, thus potentially being less reliable if not carefully tuned.
>
> We designed SimNet to have multi-headed predictions which allows significant computation wins at runtime, since a single forward-pass through the feature extraction backbone can be used across tasks. Thank you for pointing this out that it creates confusion to a reader.
> We updated the paper with the following experiments to make it clear that this is only optional for our main sim-to-real contribution and give more insight into what losses are actually important.
>
> -We first set the loss weights to zero for the 3 auxiliary output heads for the OBB prediction tasks and demonstrate that you can achieve similar accuracy with just training a single task (Table 1).
>
> -Then on the more challenging KITTI benchmark, we ablate over using the disparity loss on the output of the SCVN and find a significant drop in performance, suggesting that while the final output heads losses are not relevant to the task, the intermediate loss on disparity at the output of the SCVN is important (Table 4).
>
> >> The baseline in Section 4.1.2. is chosen by "sweeping over four different algorithms", which are not explained further in the main paper.
>
> Thank you for catching this: these are the four RGB-D variants described in Section B.1, and we have updated the text to reference this section in Section 4.1.2.
>
> >> The experimental results are presented as plain numbers without standard deviations, confidence intervals or other measures of uncertainty.
>
> We will present standard deviation statistics for the experiments by camera ready if accepted. Qualitatively, we saw that runs were very consistent across random seeds.
>
> >> Section 4.4 reports mAP@ 0.5, while previous sections report mAP@ 0.25. This choice is not further explained/justified in the paper.
>
> Thank you for pointing this out. We updated the text in Section 4.4 and Section 4.1 to add a reference to our supplemental material (Section B.3), where we discuss our evaluation method. The main reason for the discrepancy is that the metrics are measuring IOU of different structures (i.e. 2D bounding boxes vs. 3D OBBs).
>
> The 0.5 score for 2D bounding boxes on KITTI car detection is consistent with what is used in [28]. However, since OBBs can rotate freely around symmetric principal axes, we opted for a looser acceptance criteria for evaluation of 3D bounding boxes, which resulted in mAP@0.25. We note though that our grasping results indicate the predicted OBBs are accurate enough for manipulation on a physical robot.
>
> >> The appendix mentions that the weights for the losses for the different models are chosen based on HyperBand.
>
> We will add this to the main paper for final submission; thank you for the suggestion. We will also note that the need for HyperBand was specifically when using a large number of multi-headed predictions. We did not require such parameter tuning for the KITTI task, when only a single output head was used. Qualitatively, we found that the loss weights are not particularly sensitive, and in fact, we used the same loss weights when applicable across all tasks.  Please also note the additional ablation that we added in Table 1 (SimNet w/o Aux.), which also quantitatively shows that the additional tasks are not required for good OBB prediction performance.
>
> >> Contributions (iii) to (v) each refer to one experimental setup and the insights taken from it.
>
> Thank you for the suggestion.  We will add these to contribution (i).  Also, we made our final contribution to be the release of the stereo dataset and code used for evaluation. We are happy to provide an anonymized version of the code/dataset if requested.
>
> >> The RGB-D baseline in Section 4.1.2 is chosen based on the performance for "easy" objects.
>
> This is a very perceptive suggestion. We use ground truth annotations from 3D scans of easy objects to evaluate each model.  Standard RGB-D sensors make reflective or transparent objects challenging to scan, though the SCVN network does make this additional experiment feasible.  If accepted, we will run all of the networks in Section B.1 on the robot grasping task with hard objects.
>
> >> The car detection (Section 4.4) uses a "Stacked" baseline consisting of stereo images that is missing in previous sections.
>
> Thank you. We will add an ablation of the “stacked” baseline to the panoptic objects dataset in the camera ready, if accepted. We originally limited our ablation to KITTI to avoid overfitting our decision to indoor scenes dataset.

---

> > ### Comment · Reviewer_m12X · 2021-09-03
> > **Reviewer Response**
> >
> > The authors have resolved many of my concerns, while indicating that they intend to address most of the remaining ones if accepted. I can not evaluate intended changes, but still believe that the paper in its current state poses an interesting contribution. I will leave my score at `weak accept'.

---

### Official Review · Reviewer_oTrx · 2021-07-23

**Originality:** Good
**Technical Quality:** Good
**Clarity Of Presentation:** Good
**Impact:** 3

**Recommendation:**

Weak Accept: I recommend accepting the paper, but will not argue for my recommendation if the majority of other reviewers have a different opinion.

**Summary:**

The paper presents a multi-task network consisting of ResNet(-FPN) blocks and a stereo matching module that takes in stereo images and outputs different predictions such as segmentation, 3D bounding boxes, keypoints, disparity, etc. The network is trained with data generated from a non-photorealistic simulator together with domain randomization. In the robotic experiments, a T-Shirt folding task based on keypoints and unknown object grasping based on the predicted bounding boxes is tested. Furthermore, 3D/2D object detection is evaluated on human household scenes (including transparent objects) and the KITTI benchmark. They show for their network that stereo inputs outperform mono, depth or RGB-D inputs in all cases and that sim2real transfer works well.

**Issues:**

- It would be nice to see the 3D predictions from a different angle, e.g. the 3d bounding boxes in a point cloud.
- The supplement is over 50Mb, please reduce the size

**Reviewer Expertise:**

Very good: Comprehensive knowledge of the area

**Strengths And Weaknesses:**

Strengths:

- Several relevant robotic tasks are evaluated and the suitability of stereo data for these tasks is demonstrated
- Sim2Real transfer from non-realistic simulated data is shown
- Comparisons of input modalities using variations of their own network structure

Weaknesses:

- I did not learn very much. It is not a new finding that stereo can improve results when RGB-D sensors fail. Neither that disparity is a good auxiliary task for semantic predictions improving results over RGB. It is a nicely engineered system from existing components evaluated on robotic tasks. But the evaluation and discussion is lacking depth beyond "look, our system works".
KeyPose: Multi-View 3D Labeling and Keypoint Estimation for Transparent Objects, CVPR 2020
Real-Time Semantic Stereo Matching, ICRA 2020
- There is no comparison to existing approaches even when comparing on public benchmarks, e.g. on the 2d car detection of KITTI. It does not seem that results are competitive to state-of-the-art approaches regardless of modalities
http://www.cvlibs.net/datasets/kitti/eval_object.php?obj_benchmark=2d
Stereo R-CNN based 3D Object Detection for Autonomous Driving, CVPR 2019
- The reader has no idea whether the modality comparisons hold in general or whether other RGB-D methods specifically tailored for fusing modalities would turn results around.
- The argument for efficient non-realistic data generation becomes quite questionable if hyperparameter tuning takes "20 single gpu instances for 48 hours". There are easy ways to generate photorealistic annotated stereo data in much less time, e.g. BlenderProc

**Summary Of Recommendation:**

I welcome system papers and I think the approach has potential, but the evaluation is lacking quite a bit. There are no comparisons to existing approaches, no real discussions of results and not much additional findings compared to literature. Therefore, it can merely serve as a proof-of-concept.

---

> ### Author Response · Authors · 2021-08-25
> **Clarification of SimNet sim-to-real approach and comparison to prior work**
>
> We would like to thank the reviewer for their comments. We believe that there is a misunderstanding of the setting that this paper studies, which is likely our fault. We would like to clarify the problem of focus, and we have updated the paper’s introduction and text to address this. We hope the reviewer will consider these clarifications.
>
> >> It is not a new finding that stereo can improve results when RGB-D sensors fail:
>
> The papers that are listed above use real-world data and are not sim-to-real approaches.  Neither of the above approaches is trained using only synthetic data, nor have they shown a benefit of incorporating learned stereo when only synthetic data is used for training for high-level vision tasks.  So, they are not directly comparable to our main sim-to-real contribution.
>
> Our paper contributes a new technique for sim-to-real of high-level vision tasks, that relies on a stereo matching module. To our knowledge, there have been no comparisons in the literature that have shown stereo matching can enable more robust sim-to-real transfer than RGB-D based sensing for high-level vision tasks (i.e. 3D OBBs, keypoints, segmentation). The implications of such a result are: 1) optically challenging objects can be manipulated using only low-quality synthetic data and 2) there is no reliance on a physical sensor, which may have constrained supply.
>
> >> Baselines:
> There is no comparison to existing approaches even when comparing on KITTI.
>
> The existing approaches listed use real-world data, whereas SimNet is a sim-to-real approach that can be evaluated on challenging real-world tasks such as KITTI.  The results on KITTI show that the incorporation of the SCVN learned stereo network provides a scalable sim-to-real approach that is able to perform well on a range of different tasks.
>
> That said, a similar evaluation on 2D car detection for KITTI was performed in another sim-to-real approach in [28], which used mono-DR.  However, [28] did not release their code or synthetic dataset used which makes direct comparison not possible. In light of that, we did our own relative comparisons on the KITTI benchmark with our lightweight single-shot detection network. Our goal with this experiment was to illustrate the performance differences between monocular-DR, naive stereo concatenation (i.e. stacked) and a SCVN for sim-to-real transfer (See Section B.1 for the baselines that we evaluated).  We updated the experiments to show the need for the depth auxiliary loss and we added citations for the baselines that we compared against (Table 5 and Section B.1).
>
> >> The reader has no idea whether the modality comparisons hold in general
>
> We would like to thank the reviewer for their comment.  We have tried a significant number of  RGB-D permutations (four), along with monocular and depth only permutations in B.1, including replicating a similar network to Xie et al (B.1.8). Results are shown in Table 5.  If the reviewer has suggestions for comparison beyond the ones we have tried, we would be happy to discuss.  .
>
> >> The argument for efficient non-realistic data generation becomes quite questionable if hyperparameter tuning takes "20 single gpu instances for 48 hours".
>
> Thank you for your comment, and we would like to clarify the difference between loss hyperparameter tuning and dataset generation. Hyperparameter optimization was not used for data generation; the lightweight rendering presented in this paper is actually extremely cheap when compared to Blender.  Hyperparameter tuning  is used “to tune the loss weights,” as stated in section B.2, not to perform data generation. This tuning would only be needed if you wanted to use multi-head inference to reduce computation at run-time. For our KITTI experiments, we only trained a single 2D bounding box detection head, which doesn’t require hyper-parameter tuning and can be trained on a single GPU.
>
> We added an additional experiment to Table 1 that ablates the multi-task nature of the model (SimNet w/o Aux.).  This shows that multi-head training has no significant effect on the sim-to-real transfer performance of our approach.
>
> >> There are no comparisons to existing approaches.
>
> The paper is offering comparisons of our stereo transfer techniques to existing common sim-to-real techniques (i.e. monocular-DR [5], depth-based transfer [47], RGB-D combinations [4]). We updated our experiment tables to include citations where those sim-to-real baseline techniques were taken from (Table 5, Section B.1 and Section 4.1). Table 5 of the appendix provides an ablation of 8 different approaches to sim-to-real transfer on a standard stereo vision benchmark task.
>
> >> Discussions of results
>
> We added more text to the experimental results to include a better discussion of our SCVN ablation on the KITTI benchmark and our RGB-D baselines on the indoor scenes dataset. Due to space limitations, discussion was brief, but present in Section 4.

---

> > ### Comment · Reviewer_oTrx · 2021-08-30
> > **Thank you for your comments.**
> >
> > > To our knowledge, there have been no comparisons in the literature that have shown stereo matching can enable more robust sim-to-real transfer than RGB-D based sensing for high-level vision tasks (i.e. 3D OBBs, keypoints, segmentation).
> >
> > There are methods that show successful sim2real transfer for high-level stereo vision tasks that might be worth a reference, e.g.
> > [Unknown Object Segmentation from Stereo Images](https://arxiv.org/abs/2103.06796) (admittedly this one is quite recent)
> >
> > It is known that sim2real transfer for low-level vision tasks such as disparity estimation is easy and that it can benefit from high-level auxiliary tasks such as semantic segmentation. So I suppose the main contribution of this work is the observation that the other way around also holds, i.e. standard stereo matching can improve sim2real transfer for high-level vision tasks.
> >
> > > We would like to thank the reviewer for their comment. We have tried a significant number of RGB-D permutations (four), along with monocular and depth only permutations in B.1, including replicating a similar network to Xie et al (B.1.8). Results are shown in Table 5. If the reviewer has suggestions for comparison beyond the ones we have tried, we would be happy to discuss. .
> >
> > There are quite many approaches on how to fuse RGB and depth data, see for example:
> > https://arxiv.org/pdf/1808.03833.pdf , Table 17/18
> >
> > But I appreciate the ones presented in the appendix.
> >
> > > Baselines: There is no comparison to existing approaches even when comparing on KITTI
> >
> > You present a setting where you train on real KITTI data which is very important to rate the sim2real ability. It is easy and equally important to compare these real results with existing literature. I don't expect it to be SoTA, but I would like to know how mature your architecture is.
> >
> > > The argument for efficient non-realistic data generation becomes quite questionable if hyperparameter tuning takes "20 single gpu instances for 48 hours".
> > >> Hyperparameter optimization was not used for data generation;
> >
> > That is obvious, but why should you care about the efficiency of data generation if the training/tuning the networks takes so much more resources?
> >
> > > The paper is offering comparisons of our stereo transfer techniques to existing common sim-to-real techniques
> >
> > The additional experiments and ablations are definitely adding to the paper and it is great that you implemented several methods. I sincerely hope that the code will actually be open-sourced as announced because otherwise these self-implemented results without comparisons to literature are pretty much worthless.
> >
> > In conclusion, I still see a few problems with the paper. On the other hand, at least it seems to actually work well judging by the qualitative results and self-made baselines. The paper might also help to increase the popularity of stereo-based approaches for object-centric robotic vision tasks.

---

> > > ### Author Response · Authors · 2021-08-31
> > > **Thank you for your insightful comments**
> > >
> > > >> There are methods that show successful sim2real transfer for high-level stereo vision tasks that might be worth a reference, e.g. Unknown Object Segmentation from Stereo Images (admittedly this one is quite recent).
> > >
> > > Thank you for this reference, we were unaware of this work. This is a very recent preprint that we would consider submitted roughly around the time of our submission. However, we do agree that it is very relevant and will cite it in the final version of the paper.
> > >
> > > Our work and theirs offer complementary experiments to show the validity of sim2real transfer with stereo. While they consider the specific task of instance segmentation, we demonstrate that sim-to-real technique is applicable to a variety of high-level vision tasks such as 3D OBBs, keypoints and car detection.
> > >
> > > Furthermore, we demonstrate that the technique works well with very low quality datasets, whereas their approach is tested with photo-realistic scenes from SUNCG. Finally, our work demonstrates that this technique is reliable enough to enable robust manipulation of glassware in multiple homes.
> > > We believe these additional experiments complement their work and together these two papers offer strong evidence for stereo based sim-to-real transfer.
> > >
> > > >> There are quite many approaches on how to fuse RGB and depth data, see for example: https://arxiv.org/pdf/1808.03833.pdf , Table 17/18 … But I appreciate the ones presented in the appendix.
> > >
> > > Thank you for the comment.  We hope that this paper provides value to the community in the comparisons that were performed and do see that there are a handful of additional variants that we could compare to.  We will try to compare against more of the variants for the final version of our paper.
> > >
> > > >> You present a setting where you train on real KITTI data which is very important to rate the sim2real ability. It is easy and equally important to compare these real results with existing literature. I don't expect it to be SoTA, but I would like to know how mature your architecture is.
> > >
> > > Thank you for your comment.  Although the main reason for the KITTI experiments is to understand how close our sim-to-real approach would come to an approach trained on real data with the same method, we do see your point. For the final version of the paper, we will report SOTA numbers for similar single-shot network architectures for relative comparison.
> > >
> > > >> That is obvious, but why should you care about the efficiency of data generation if the training/tuning the networks takes so much more resources?
> > >
> > > Thank you for your comment.  To clarify, as shown by the ablation of all but the OBB output head in Table 1, the multi-task hyper-parameter search is optional and only needed for efficient runtime inference of many tasks.
> > >
> > > Low quality rendering can save computation resources, but also has the added benefit of not needing artists to create photo-realistic scenes. For example, datasets like SUNCG require expensive artist annotation, which makes it difficult to create data for new domains. By having such low-quality scene layout and texture randomizations, it's relatively easy for SimNet to switch from car detection to indoor scene understanding.
> > >
> > > .>> The additional experiments and ablations are definitely adding to the paper and it is great that you implemented several methods. I sincerely hope that the code will actually be open-sourced as announced because otherwise these self-implemented results without comparisons to literature are pretty much worthless.
> > >
> > > We are happy that the additional results helped.  To clarify one point, the code and dataset have already been published publicly on github and have received multiple downloads from other researchers. Unfortunately, it has been proven difficult to share without breaking anonymity, since some of the scenes have traceable markers in them.

---

> > > > ### Comment · Reviewer_oTrx · 2021-09-01
> > > > **Raise of score**
> > > >
> > > > Since the authors have resolved many of my issues and concerns, I have raised the score to "weak accept".

---

### Official Review · Reviewer_o8ha · 2021-07-23

**Originality:** Excellent
**Technical Quality:** Excellent
**Clarity Of Presentation:** Very Good
**Impact:** 4

**Recommendation:**

Strong Accept: I recommend accepting the paper and will argue for my recommendation even if other reviewers hold a different opinion.

**Summary:**

This paper presents an architecture to predict object segmentation, oriented bounding boxes, keypoints and disparity from stereo RGB images provided as input. The training data is purely synthetic, generated in simulation via domain randomization. The trained system enables manipulation of objects in the real world, including transparent and reflective objects as well as non-rigid objects such as t-shirts. It can also achieve 2D car detection, as shown in experiments involving the KITTI benchmark.

**Issues:**

Figure 2: "The output of the SCVN is a low resolution disparity image fed in with features from the left image to a ResNet-FPN backbone and output prediction heads" --> can the authors please better explain what are the features from the left image? How are they obtained?

Can the authors please explain the dimensionalities Wo/8 X Ho/8 X 16, Wo/8 X Ho/8 and Wo/8 X Ho/8 X 6? This is in subsection "Oriented Boundin Boxes". It's not clear to this reviewer why the oriented bounding boxes prediction leads to these shapes.

Typos:

- The second component and a core contribution SimNet (Figure 2) --> contribution of
- However, the robot is only able grasp 35% of the “hard” objects with the RGB-D network --> able to grasp
- we can learn robust “low-level” features like disparity than enable sim2real transfer of “high-level” vision tasks --> that enable
- Each prediction head high uses the up-scaling branch defined in [40] --> What is meant here by "high"?
- Since our SCVN, produces a disparity image at quarter resolution --> no comma between "SCVN" and "produces"


**Reviewer Expertise:**

Fair: Some knowledge of the area

**Strengths And Weaknesses:**

Strengths:
- Surprising performance in manipulation of transparent, reflective and non-rigid objects.

Weaknesses:
- A few elements can be better explained to increase the clarity of the paper.

**Summary Of Recommendation:**

This work achieves remarkable results in manipulating challenging objects (transparent, reflective, non-rigid).

The release of their dataset is also a significant contribution.

---

> ### Author Response · Authors · 2021-08-25
> **Clarification of SimNet Model Features and OBB Dimensions**
>
> We would like to thank the reviewer for his positive feedback and suggestions on typos.  We have already addressed some of them and will make sure they are all addressed in the final version.  We address the additional comments below:
>
> >> Figure 2: "The output of the SCVN is a low resolution disparity image fed in with features from the left image to a ResNet-FPN backbone and output prediction heads" --> can the authors please better explain what are the features from the left image? How are they obtained?
>
> Thanks for pointing this out, and we will clarify this further in the paper.  The features are from a standard ResNet Stem which outputs 32 channels that are 4x downsampled from the original image. We will add more detail to the supplemental material in the final version.
>
> >> Can the authors please explain the dimensionalities Wo/8 X Ho/8 X 16, Wo/8 X Ho/8 and Wo/8 X Ho/8 X 6? This is in subsection "Oriented Bounding Boxes". It's not clear to this reviewer why the oriented bounding boxes prediction leads to these shapes.
>
> The OBB head outputs a set of image sized tensors, which are then indexed to create the parameterized 3D structure. We plan to release the code for the camera ready, which will hopefully provide more information of the network structure.

---

> ### Comment · Reviewer_o8ha · 2021-09-01
> **Reviewer Response**
>
> Thank you for answering to the issues raised by the reviewers. In my opinion, the paper presents a significant contribution to the robotics community especially when it comes to manipulating non-rigid, transparent and reflective objects. My score remains unchanged.

---

### Official Review · Reviewer_z3Gb · 2021-07-26

**Originality:** Good
**Technical Quality:** Fair
**Clarity Of Presentation:** Very Good
**Impact:** 2

**Recommendation:**

Weak Reject: I recommend rejecting the paper, but will not argue for my recommendation if the majority of other reviewers have a different opinion.

**Summary:**

This paper proposes a perception model, SimNet, that leverages learned stereo matching to predict keypoints, oriented bounding boxes, and segmentation while only using low-quality simulation data during training. The authors perform extensive experiments in both indoor and outdoor environments and show that the proposed method can achieve better performance compared to monocular, depth-only, or RGB-D input.


**Issues:**

**Major**:
- Show results to justify why designing SCVN is necessary. For example, does it perform better than published learned stereo matchers for disparity prediction? Does it lead to better downstream tasks performances compared to published learned stereo matchers?
- Show ablation study of the disparity auxiliary loss (line 169-174).
- If there are two-stage approaches that predict “3D oriented bounding boxes of unknown objects” compare SimNet’s accuracy, runtime, and memory consumption to them.
- Show disparity prediction and segmentation results on KITTI. Compare the results to other published learning-based methods for readers to understand how does SimNet compared to others.
- Study whether the multi-task nature of SimNet helps/hurts the performance.
- Compare the segmentation results to Xie et al. [6].

**Minor**:
- Line 264, Figure 6 only exists in the supplementary material but authors didn’t mention it.
- Line 247, “different, real homes” -> “different real homes”.
- In Figure 4 & 5, both the depth and the disparity are too small for readers to scrutinize. I think showing less examples while explicitly pointing out why SimNet is better would help readers a lot.


**Reviewer Expertise:**

Very good: Comprehensive knowledge of the area

**Strengths And Weaknesses:**

**Strengths**:
- The motivation is clear and sound.
- The technical presentation is clear.
- The experiments on the real world data of different domains show that the proposed method is versatile.

**Weaknesses**:
- To me, the major contribution of the paper is unclear.
  - The authors didn’t explain why they had to design Stereo Cost Volume Networks (SCVN, Section 3.1). Is it a better learned stereo matcher compared to existing works listed in Section 2.3?  Can authors replace it with an existing work? If so, does the performance of other heads (e.g., keypoints, segmentation) change?
  - How important is the disparity auxiliary loss (line 169-174)? The authors didn’t ablate SimNet to show its importance. In the supplementary material, Table 5 seems to indicate that incorporating this loss doesn’t improve other baselines much.
  - The second contribution indicates “the first network to enable single-shot prediction of 3D oriented bounding boxes of unknown objects”. Are there two-stage approaches? If so, how do SimNet’s accuracy, runtime, and memory consumption compare to them?
  - Line 71 “(iii) stereo vision experiments on simulated and real home scenes with annotations for oriented bounding boxes, segmentation masks, depth images, and keypoints” -> I don’t think running experiments on these known tasks itself is a contribution.

- Experiments:
  - Line 72 indicates that one of the contribution is “(iv) experiments on the KITTI stereo vision benchmark dataset that suggest that the network transfers significantly better than standard domain randomization and naive stereo concatenation” -> How does it compare to other state-of-the-arts (SOTAs) for 2D car detection that also report results on the same benchmark? Additionally, reporting results on other tasks will help readers understand the potentials and limitations of SimNet. As SimNet can perform segmentation and disparity prediction too, can authors show results on those tasks and compare to published methods on KITTI?
  - As SimNet is a multi-task framework, do these tasks help each other? I think the authors can ablate the tasks used to train SimNet to inform readers whether multi-task learning helps or hurts the performance in this case.
   - Line 108 the authors mentioned that the approach is comparable to Xie et al. [6], can you show the compared performance on segmentation?


**Summary Of Recommendation:**

I recommend rejecting the paper because 1) the ambiguous contribution of the paper and 2) the lack of experiments (see my comments above). I think authors need to perform a series of ablation studies to justify the design of the pipeline. Additionally, comparing SimNet to published works on famous benchmarks (e.g., KITTI) for tasks such as segmentation and disparity is important for readers to understand how mature SimNet is. Lastly, an ablation study on the number of downstream tasks to train on is important for readers to understand how it affects the peroformance.

---

> ### Author Response · Authors · 2021-08-25
> **Clarification of Approach and Additional Experiments**
>
> >> The authors didn’t explain why they had to design Stereo Cost Volume Networks (SCVN, Section 3.1).
>
> The focus of this work is on approaches that work well for sim-to-real transfer and not achieving  highly accurate disparity predictions. In light of this, we tried to examine an architecture that enabled robust transfer but didn’t require computationally expensive operations such as 4D feature volumes or 3D convolutional operations.
>
> We found that SCVN can enable reliable extraction of geometric features via explicit feature comparisons across the scan-line. Naive concatenation of left and right stereo images wasn’t as robust for transfer in our ablation on KITTI. In the Appendix, we provide an evaluation of 8 additional sim-to-real techniques (Table 5), and in this revision we have additionally added two more ablations of the SCVN architecture in Tables 1 and 4.
>
> >> How important is the disparity auxiliary loss (line 169-174)?
>
> This is a great suggestion, and we have updated Table 4 to include this ablation for SimNet + SCVNs. Please see the SCVN ablations on the KITTI benchmark.  The auxiliary loss was very important for high accuracy results. The auxiliary loss seems essential when trying to learn geometric features. Regarding Table 5, we use a depth auxiliary loss on non-SimNet architectures, however since the geometry isn’t being learned the refinement step doesn’t seem to help transfer.
>
> >> The second contribution indicates “the first network to enable single-shot prediction of 3D oriented bounding boxes of unknown objects”.
>
> We believe that SimNet is the first to predict OBBs directly from the network output. An alternative approach to direct OBB prediction is to segment out the object with an instance mask detector and then to compute PCA on the resulting point cloud for an object. However, the two-stage approach is an approximation, since it can’t reason about the whole object (e.g., of the back of the object, which is unobserved) and therefore the OBBs have inaccurate depth.
>
> In the paper we show a simple modification to an existing absolute pose loss (see MobilePose [41]) which can enable OBB prediction of objects, We consider this a minor contribution of the paper compared to our proposed sim-to-real technique.
>
> >> Line 72 indicates that one of the contribution is “(iv) experiments on the KITTI stereo vision benchmark dataset that suggest that the network transfers significantly better than standard domain randomization and naive stereo concatenation”
>
> Thank you for pointing this out. The aim of this paper is to present a sim-to-real approach, and to only use synthetic data for training.  The traditional KITTI benchmarks use real-world training data and are not focused, as this paper is, on sim-to-real transfer.  Other comparable approaches that do sim-to-real transfer [28] have evaluated their sim-to-real approaches on the 2-D car prediction component of KITTI, which is what we did in this paper.  Although they did not release code or data to compare against, the monocular result in Table 4 is a similar sim-to-real technique as that in [28].
>
> We are not claiming that our approach would beat state-of-the-art using real-world data on KITTI. We used this experiment as an ablation over our SVCN sim-to-real technique to illustrate the performance improvements of our transfer algorithm relative to other sim-to-real variants (i.e. naive concatenation and monocular DR).
>
> >> As SimNet is a multi-task framework, do these tasks help each other?
>
> This is a perceptive comment, and we have now added an additional experiment to study this. Our aim in creating a single multi-task framework is to enable the addition of new outputs easily and scalably. In this revision, we have modified the paper to describe the segmentation and full-res disparity prediction as optional heads (Section 3.3) and added an additional ablation study in Table 1 on our indoor scene dataset.
>
> We found no significant performance difference (i.e. <0.03 mAP) when training without all the auxiliary losses, but still including the standard auxiliary disparity loss.  As a result, we see that single task learning is just as effective.
> We apologize for the confusion it caused having them in our paper. Our aim was to showcase the ease of multi-head training with densely annotated synthetic datasets, however it's clear that caused significant confusion.
>
> >> Line 108 the authors mentioned that the approach is comparable to Xie et al. [4], can you show the compared performance on segmentation?
>
> As stated for the meta-reviewer, a two-shot sim-to-real approach similar to Xie et al was evaluated in Section B.1.8 and the results are in Table 5 (Appendix).  Table 1 also now has a comparative result to Xie et al (RGB-D Seq).  The evaluation is on the OBB prediction head for our indoor scene dataset. A direct comparison is not possible because the instance mask segmentation datasets used in [4] didn’t have stereo data.

---

> > ### Comment · Reviewer_z3Gb · 2021-09-03
> > **Reviewer Response**
> >
> > Hello, thank you for the clarification!
> >
> > I am still concerned about the contribution with respect to the model architecture.
> >
> > > In light of this, we tried to examine an architecture that enabled robust transfer but didn’t require computationally expensive operations such as 4D feature volumes or 3D convolutional operations.
> >
> > To support this claim, I think authors need to try existing model architectures for the stereo matching and report accuracy/speed trade-off before designing a new model architecture. As this is claimed to be the major contribution of the paper, I will keep my original score.

---

> > > ### Author Response · Authors · 2021-09-09
> > > **Clarification on Stereo Architecture**
> > >
> > > We thank the reviewer for their final opinion.
> > >
> > > To clarify a point, our choice in stereo architecture was a standard stereo network architecture. It is based on DispNet-C[15], which consists of two feature extractors fed into a 1D correlation cost-volume (i.e. SVCN) which is processed by a series of convolutions and then predicts disparity. Given that DispNet-C was published in 2016, we chose to update the architecture with more modern feature extractors (i.e. ResNets), which is commonly supported in training libraries (e.g. pytorch, tensorflow).  We apologize for the confusion and will try to update the paper to make it more clear where our architecture came from.
> > >
> > > Our main contribution is showing the addition of a stereo network without “bells and whistles”' can enable robust sim-to-real transfer of high-level vision tasks from low quality synthetic data. We feel this is a significant contribution because it demonstrates a very promising way to approach  sim-to-real transfer compared to RGB-D baselines.
> > >
> > > However, we agree with the reviewer that this might not be the optimal way to learn geometric features for transfer and other stereo networks should be considered with respect to speed/accuracy tradeoffs. In light of this, we hope that by releasing our indoor scenes stereo datasets and training code, more research into this topic can be explored by the community.

---

### Meta-Review · Area_Chair_1JXs · 2021-08-12

**Recommendation:** Accept (Poster)
**Confidence:** 4

**Metareview:**

On one side, the reviewers agree that the experimental results presented in the paper are notable, giving good evidence that the proposed approach based on the SimNet architecture is useful and effective. On the other side, there are different perspectives regarding the scientific contribution of the work. In any case, a clearer distinction of the proposed algorithm to existing work (e.g. Xie et al [6], Liu et al (see reviewer oTrx)) and a more thorough motivation for the design of this new method are necessary to assess the contribution. What is it that the SimNet architecture can do that can not be done by similar ones? Also, the ablation studies mentioned by reviewer z3Gb are an important add-on to measure the impact of the approach. Auxiliary tasks are used more and more often, and intuitively, it is reasonable to use them, but an empirical evidence of their benefit is still required. It would be good if the authors could react on these two main points in their answer to gain more insight about the main contributions of their work.

Post-rebuttal:
During the discussion phase, the authors could clarify most of the concerns raised by the reviewers. The additional results and the added ablation studies are very useful, showing the usefulness and significance of the approach. Also, the further distiction from existing works is convincing.

---

> ### Author Response · Authors · 2021-08-25
> **Clarification of Contributions and Comparison to Prior Work**
>
> We  thank the meta-reviewer for their review. We address the concerns below.
>
> >> There are different perspectives regarding the scientific contribution:
>
> Thank you for pointing this out, we understand that we could improve clarity of the contribution based on the feedback from the reviewers. We have updated the introduction of this paper to reflect the main contribution of the paper. The main contribution of the paper, as stated in the final paragraph of the introduction on page 2, is:
> An efficient neural network… that leverages learned stereo matching… using low-quality synthetic data to transfer from simulation to reality.
>
> Although stated indirectly here, the main contribution is our sim-to-real approach, which uses learned stereo matching to transfer high-level vision tasks from simulation. SOTA sim-to-real techniques in robotics (e.g., Xie et al) require the use of active RGB-D sensors (e.g. Realsense, Kinect), which can limit performance to scenes with matte objects and low natural light.  These sensors also have limited availability due to supply chain issues.
>
> Our sim-to-real approach, SimNet, demonstrates an alternative to the use of RGB-D sensors that uses a learned stereo matching network that is good enough to perform manipulation tasks and doesn’t face the above limitations. Our experiments are designed to test this new sim-to-real technique on OBB prediction and not on benchmarks for other real-world downstream tasks (i.e. segmentation, disparity prediction, 2D detection). We updated our introduction of the paper to address this.
>
>
> >> A clearer distinction of the proposed algorithm to existing work (e.g. Xie et al [6], Liu et al (see reviewer oTrx)).
>
> Thank you for asking this clarification question. We will briefly describe the notable differences between this work and the mentioned papers below.
>
> >> Comparison to Xie et al:
>
> In Xie et al, the authors propose both a sim-to-real technique, with RGB-D sensors, and a method for instance segmentation. A direct comparison on the downstream task of instance segmentation was not possible due to the lack of the availability of an indoor scene dataset with stereo images.  However, a comparison to their proposed sim-to-real technique was performed to the best of our abilities in Appendix (B.1.8) and can be seen in Table 5.  The corresponding RGB-D Seq. baseline was added to Table 1 and the discussion of Section 4.1.
>
> >> Comparison to Liu et al:
>
> Liu et al. studies the problem of learning keypoints on transparent objects given access to real-world training data from a custom data collection rig.  Since the core research question of Liu et al does not involve sim-to-real transfer, it is quite different from our paper, whose primary research question is how to enable robust sim-to-real transfer from low-quality synthetic data.  We cited this paper in the introduction of our paper to illustrate alternative approaches to the perception of glassware that don’t require low-quality synthetic data but instead use real-world data collection.
>
> >> A more thorough motivation for the design of this new method is necessary to assess the contribution.
>
> We would like to thank the meta-reviewer for this question.  From the reviews we can see that the focus on sim-to-real transfer was not as clearly communicated as we would have liked, and the incorporation of the SCVN was not sufficiently motivated by ablations. The paper has been updated with more motivation for the SCVN and why it enables robust sim-to-real transfer, especially in the introduction and experiments. Additional ablation studies were added, including on the KITTI dataset, to motivate our design decisions (Table 1 and Table 4).
>
> >> What is it that the SimNet architecture can do that can not be done by similar ones?
>
> SimNet differentiates itself from other approaches in that it is able to perform sim-to-real transfer of a variety of high-level vision tasks, and which is made robust by incorporating the SCVN learned depth network.  By learning coarse geometric features, it can enable robust sim-to-real transfer when compared to existing techniques (Table 5).  This specifically enables training on low-quality synthetic data and being able to manipulate optically challenging objects such as glassware.
>
> >> Also, the ablation studies mentioned by reviewer z3Gb are an important add-on to measure the impact of the approach. Auxiliary tasks are used more and more often…, but an empirical evidence of their benefit is still required.
>
> These additional experiments have been added to the paper as a part of the KITTI benchmark. The auxiliary loss on the SCVN output is very important for producing accurate bounding boxes in the KITTI domain (Table 4).  However, the single multi-task model is not important to make the different output heads more accurate (Table 1), but does provide runtime benefits in terms of GPU memory and compute usage. Please refer to Tables 1 and 4 for the new ablation experiments.

---

### Decision · Program_Chairs · 2021-09-13

**Decision:**

Accept (Poster)

**Comment:**

On one side, the reviewers agree that the experimental results presented in the paper are notable, giving good evidence that the proposed approach based on the SimNet architecture is useful and effective. On the other side, there are different perspectives regarding the scientific contribution of the work. In any case, a clearer distinction of the proposed algorithm to existing work (e.g. Xie et al [6], Liu et al (see reviewer oTrx)) and a more thorough motivation for the design of this new method are necessary to assess the contribution. What is it that the SimNet architecture can do that can not be done by similar ones? Also, the ablation studies mentioned by reviewer z3Gb are an important add-on to measure the impact of the approach. Auxiliary tasks are used more and more often, and intuitively, it is reasonable to use them, but an empirical evidence of their benefit is still required. It would be good if the authors could react on these two main points in their answer to gain more insight about the main contributions of their work.

Post-rebuttal:
During the discussion phase, the authors could clarify most of the concerns raised by the reviewers. The additional results and the added ablation studies are very useful, showing the usefulness and significance of the approach. Also, the further distiction from existing works is convincing.